# A Fine-Grained Modeling Approach for Systolic Array-Based Accelerator

Yuhang Li , Mei Wen *, Jiawei Fei, Junzhong Shen and Yasong Cao

School of Computer Science, National University of Defense Technology, Changsha 410073, China
* Correspondence: meiwen@nudt.edu.cn

**Abstract:** The systolic array provides extremely high efficiency for running matrix multiplication and is one of the mainstream architectures of today's deep learning accelerators. In order to develop efficient accelerators, people usually employ simulators to make design trade-offs. However, current simulators suffer from coarse-grained modeling methods and ideal assumptions, which limits their ability to describe structural characteristics of systolic arrays. In addition, they do not support the exploration of microarchitecture. This paper presents FG-SIM, a fine-grained modeling approach for evaluating systolic array accelerators by using an event-driven method. FG-SIM can obtain accurate results and provide the best mapping scheme for different workloads due to its fine-grained modeling technique and deny of ideal assumption. Experimental results show that FG-SIM plays a significant role in design trade-offs and outperforms state-of-the-art simulators, with an accuracy of more than 95%.

**Keywords:** modeling; systolic array; accelerator

## 1. Introduction

Deep neural networks (DNNs) have come to play an increasingly significant role in image recognition, speech recognition, text classification and other fields [1–3]. As application requirements have continued to increase, complex network models and a large numbers of parameters have resulted in higher computation time and declining performance. Deploying hardware accelerators is a common way for people to solve such problems. These accelerators [4–10] use different dataflows and hardware architectures to accelerate the computation in different ways. Existing work shows that dataflow in particular has a substantial impact on data reuse and hardware utilization [7,11,12].

With the goal of finding efficient dataflows and hardware architectures, many previous works [7,13–18] have attempted to explore the design space with varying degrees of success. The systolic array has also become one of the mainstream DNN accelerator architectures due to its unique structural characteristics. However, in order to improve the generality of the models, these works [16–18] are modeled in an overly abstract way, making it difficult to fully describe the specific implementation details of the hardware, and the simulation results obtained are far from the real situation. In addition, these models [15–18] are established based on certain assumptions, such as a lack of correlation between data, sufficient data supply during computations, etc. These assumptions are often difficult to implement in practice, and the resulting problems are difficult to capture in the model. As a result, the pauses caused by these problems can also lead to inaccurate simulation results. On the other hand, when the dataflow and hardware structure have been predetermined, the hardware will also employ different scheduling operations and data segmentation during computations, which we refer to as different mappings. Since these different mappings can have a significant impact on performance, it is also very important to quickly search for and identify the best mapping for different workloads under given conditions. Much of the previous work [17,18] has not considered this issue and thus has not been able

to evaluate different architectures fairly enough. Timeloop [16] considered this problem but does not support affine functional simulation and therefore cannot support systolic arrays.

To address these challenges, we propose FG-SIM, a fine-grained method for systolic array-based accelerator evaluation, and implement a systolic array-based accelerator simulator. This method describes the details of hardware implementation, simulates the influence of hardware resource constraints, data dependence and other factors on the calculation process, and further improves the accuracy of the simulation results. In summary, this work makes the following contributions:

- We propose a fine-grained method for evaluating systolic array accelerators. Through the use of fine-grained modeling methods, we achieve the description of mapping, affine, pipeline, data dependence and hardware resource dependence such that the obtained results are close to the real architecture.
- We construct an event-driven simulator, which is centered on computing and is suitable for exploring the microarchitecture under the systolic array structure. It can find the optimal mapping for different workloads. Compared with a cycle-accurate simulator, it achieves faster simulation and can support larger-scale parameters.
- We use computation time to quantify performance, and the results show that our method can more accurately describe and predict the relevant performance of the simulator. Compared to other known works, the accuracy of our method is far higher, with values above 95%.

## 2. Motivation

Due to the increasing demand for computing, the application of accelerators has become more widespread in recent years. This has prompted many accelerator-related works to be proposed and realized, such as TENET [17] and MAESTRO [15]. These efforts typically use simulators to evaluate different architectures in order to identify those that are most efficient. In order to enable the simulator to evaluate a variety of architectures, the above works focused on improving the generality of the model. However, while the coarse-grained modeling approach improves generality, it ignores many details of hardware implementation. At the same time, the excessive pursuit of generality introduces further conditional assumptions into the modeling that often do not hold in practice; thus, the results obtained by the model are not accurate.

On the other hand, common DNN accelerators only expose the range of available and configurable settings in the hardware. The behavior of the accelerator is closely related to these configurations; different configurations correspond to different mapping methods and computation processes, and the final computation times also differ significantly. Timeloop [16] shows the energy efficiency distribution of convolution computations for various mappings on a 1024-MAC architecture similar to NVDLA [19], and the results also support this conclusion. The reason for this phenomenon is that different mapping schemes represent different partitioning and scheduling schemes. Efficient mapping can maximize reuse and reduce computation time. Figure 1 illustrates the time required to multiply two $256 \times 256$ matrices with different mapping methods under the architecture of Figure 2. The size of the systolic array is $16 \times 16$, the size of each buffer on the chip is 512 KB, and the bandwidth is 512 bits.

In addition, for the DNN network, the performance of different layers may vary substantially due to their dimensions and flattening. Therefore, evaluating any individual neural network layer alone cannot represent the performance of the entire network. Since the optimal mapping for each layer of the network may also be different, it is also necessary to find the corresponding optimal mapping according to the characteristics of different neural network layers if we are to evaluate the overall performance of the network as objectively as possible. Table 1 shows four evaluations of accelerator modeling methods. Timeloop fully considers the important impact of different mapping schemes on performance. Unfortunately, due to the limitations of the description method, Timeloop cannot support complex data streams with skewed data access. TENET, as a relatively advanced

modeling method (at present), is better able to compensate for the deficiencies of the above two schemes; however, its modeling does not support pipelines. SCALE-Sim is a simulator for accelerators with systolic array structures, which focuses more on discussing the impact of systolic array shape and size on computational efficiency. It is not sensitive to different datastreams when estimating computation time, and the constructed mapping space is not comprehensive. In addition, the above work relies on ideal assumptions for modeling, ignoring the adverse effects of data dependence and hardware resource dependence on computation.

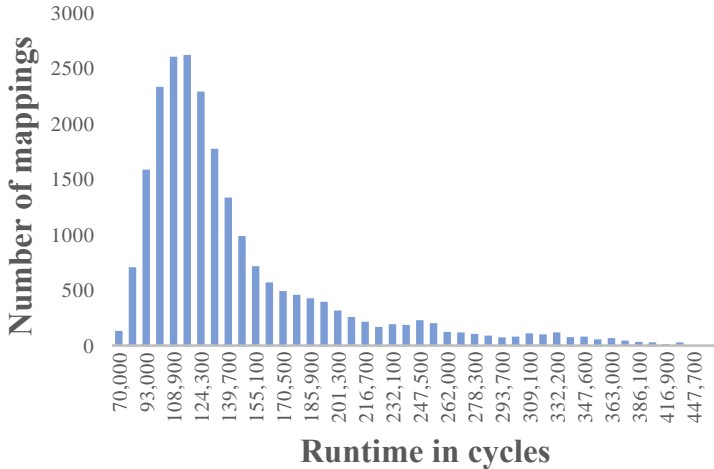

**Figure 1.** Performance comparison of different mapping schemes.

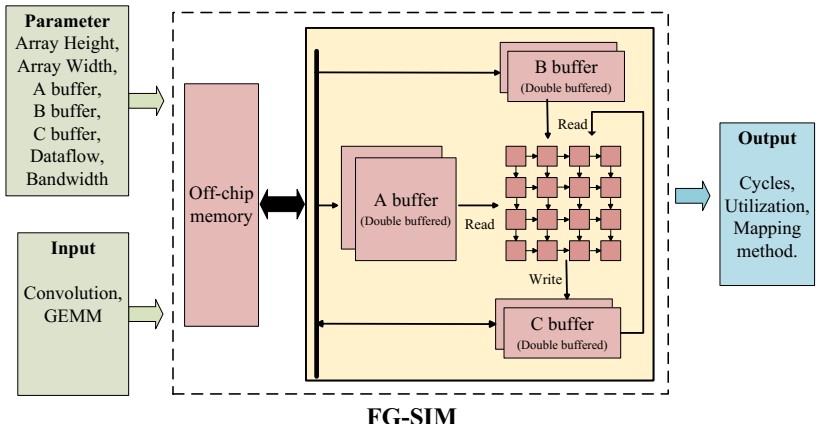

**Figure 2.** Schematic depicting FG-SIM, with inputs and outputs.

**Table 1.** Comparison of Four Modeling Methods of Accelerator.

|  | Timeloop [16] | TENET [17] | FG-SIM | SCALE-SIM [18] |
|---|---|---|---|---|
| Affine loop transformation | × | √ | √ | √ |
| Pipeline | × | × | √ | × |
| Data dependence | × | × | √ | × |
| Hardware dependence | × | × | √ | × |
| Data assignment analysis | × | √ | √ | √ |
| Precise reuse analysis | × | √ | √ | √ |
| Mapping space exploration | √ | × | √ | √ |

Considering the above issues, we aim to adopt a fine-grained approach to constructing the accelerator model and build a simulator that is closely integrated with the hardware accelerator on this basis. Through the use of more fine-grained description, it can simulate many details in the hardware implementation and obtain more accurate simulation results. At the same time, we hope to be able to implement a mapper based on the same principles. By quickly finding an effective mapping, we can make fuller use of the hardware performance.

## 3. Related Work

In recent years, a large number of excellent works [7,20–24] have been published on the subject of DNN accelerators. In order to maximize the versatility of the work, they need to describe the organization of the different designs. However, this also results in their inability to provide a relatively complete and detailed cost model of the architecture [25,26]; alternatively, it may be difficult to accurately infer the reuse of space or time in the computation process due to the simple hardware template.

MAESTRO [15] is a data-centric analysis cost model that is used to estimate the various cost-benefit trade-offs of data streams. It introduces a set of data-centric directives that are used for analysis to obtain various forms of reuse and exploit them using hardware capabilities. However, it supports only a limited variety of data streams and cannot support hardware data streams following affine transformation (e.g., data streams of 2D systolic arrays).

TENET [17] is a relation-centric simulator that employs the polyhedron model library ISL. Compared with the polynomial model, its computation accuracy is higher. However, the assumptions made by the model are too ideal; it assumes that all correlations are ignored, that data computation and transmission are completely streamlined and that the numbers of cycles are the same. Thus, TENET computations also produce errors in reality.

Timeloop [16] is a computation-centric infrastructure designed to evaluate and explore the architecture design space of deep neural network accelerators. It describes the design space using a concise and unified loop representation with mapping directives. However, it is unable to perform affine functional simulation and cannot support systolic array structure.

SCALE-Sim [18] is a configurable systolic array-based cycle accurate DNN accelerator simulator. It exposes various micro-architecture features to designers and achieves comprehensive design space exploration on this basis. However, it assumes that no dependencies exist between the data and that the data supply is adequate during the computation process. This assumption does not truly reflect the computation situation, meaning that the results obtained are insufficiently accurate.

## 4. FG-SIM Overview

Figure 2 shows the schematic representation of FG-SIM along with an example of its inputs and outputs. The accelerator mainly consists of a systolic array and several on-chip buffers. A systolic array is a regular array of multiply-accumulate units that achieves efficient computation by leveraging data reuse within the array.

The computing-centric simulator we built can create a mapping space for a given workload, and can also obtain optimal accelerator computing performance under any mapping. The core concept of our modeling is to use a fine-grained way to tile. Utilizing the fact that the computation and data movement patterns in DNN computations are largely deterministic, we can use the size of the on-chip buffer as a restriction to divide the data into tiles. This modeling method is beneficial to clearly describe the data interaction process between off-chip storage and on-chip buffers. For the calculation process within each tile, we again use the tile concept. Specifically, we use the size of the systolic array as a constraint and divide each tile into multiple sub-tiles. This method of secondary division can describe the calculation process in a more fine-grained manner, enabling us to discover

the adverse effects of data dependence, pipeline bubbles and other issues on performance, and thereby greatly improve the model accuracy.

Figure 3 illustrates the basic workflow of the simulator. After our simulator receives the input, it divides the data into tiles according to the mapping scheme. This corresponds to the process of fetching data from off-chip memory to on-chip buffers in Figure 2. For each tile, the simulator will perform calculations based on the size and shape of the tile to obtain the computation time corresponding to the tile. This corresponds to the process in Figure 2, where the data from the on-chip buffer are involved in the computation. On this basis, the model analyzes the tiles in terms of both data dependence and hardware resource dependence, and finally obtains the final overall computation time. In more detail, our simulator requires the following input:

- The shape and parameterization of the workload (for example, the relevant data of the input, output and weight tensor in the convolutional layer, or the parameters of the matrix in the GEMM).
- The hardware organization of the architecture (on-chip storage scale, bandwidth).
- The mapping method.

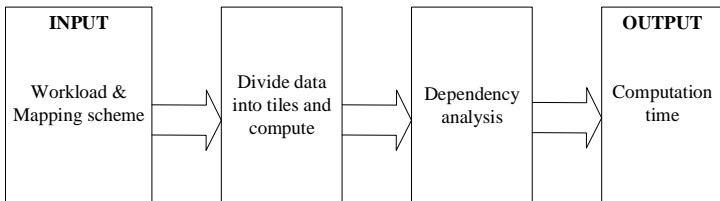

**Figure 3.** The basic workflow of FG-SIM.

In addition, we also built a mapper based on the simulator. Based on the scale of the input data and the amount of available hardware resources, the mapper constructs a mapping space for the workload. According to the size of the data, we can employ traversal or heuristic methods to quickly identify the optimal mapping that will give the greatest boost to the hardware performance.

**5. FG-SIM Simulator**

Our simulator is an event-driven simulator written in C++. It simulates the process of moving data from off-chip storage to on-chip buffers, performing computations, obtaining results and then storing them back in off-chip storage. It generates a series of operations based on the hardware data flow and determines the dependencies between the operations. In addition, it also performs high-level optimizations, such as software pipelining.

In more detail, when the user specifies the workload and mapping method, the simulator will automatically divide the data into tiles according to the mapping method. Due to the deterministic nature of the computation process, once the workload and mapping method are determined, we can easily calculate the size of each tile and its corresponding serial number using algebra. This step plays a significant role in subsequent computations and dependency simulations. After the adverse effects of memory access conflicts, computation delays and other factors have been fully considered, the simulator will output the computation time and data movement time corresponding to each tile. Finally, after considering data dependency and hardware resource constraints, we can obtain the final simulation computation time.

*5.1. Simulation of Affine Function*

In order to efficiently utilize the computing performance of the systolic array, it is necessary to ensure the horizontal and vertical transmission of data in the systolic array during the computation process. Given that the time required to compute the processing element (PE) and the time required to transmit the data may not be the same, researchers often employ an affine means of description. Many previous works did not support this

function, meaning that the structure of 2D systolic arrays could not be properly simulated. Although the most advanced TENET supports the affine function, it also assumes that the time required for PE computation and data transfer is the same, and cannot describe the situation in which these two times are different. In addition, TENET is unable to adequately describe the pipeline optimization in the computation process; in some cases, the estimated performance obtained differs substantially from the real situation.

Our simulator supports the affine function of 2D systolic arrays. In order to describe the model in a more fine-grained manner, we introduce the computation delay to describe the PE computation time. When modeling, we assume that it takes one cycle for data to be transmitted between PEs, while the number of cycles required for data to be calculated is the computation delay. When data are transferred into the systolic array to aid in performing the computation, the data of the same row are transferred in sequence at specific intervals in a cycle, and the data of adjacent rows are transferred to the systolic array in sequence at an interval dictated by the computation delay. On this basis, we also realize the optimization of the pipeline, which further improves the accuracy of the simulation.

For example, Figure 4 describes the situation when the computation delay is two cycles. We use (x, y) to represent the serial number of each data point. x represents the number of rows in which the data is located, while y represents the order of the data in this row. When the computation begins, the data corresponding to the point (1, 1) are first transmitted laterally into the systolic array and participates in the computation. In the next cycle, the data of (1, 2) are transferred to the systolic array. When the total time reaches the third cycle, the data corresponding to (1, 1) have already been calculated and can be passed on as the result. Therefore, the data of (2, 1) also begin to be passed laterally into the systolic array, and participate in the new computation together with the computation result of the (1, 1) data. This method ensures that the horizontal transmission and vertical multiplexing of data are carried out simultaneously. By analogy, the data of each row can be transmitted to the systolic array only after waiting for two cycles after the data of the previous row are transferred to the systolic array. In this way, we can realize the corresponding affine function simulation.

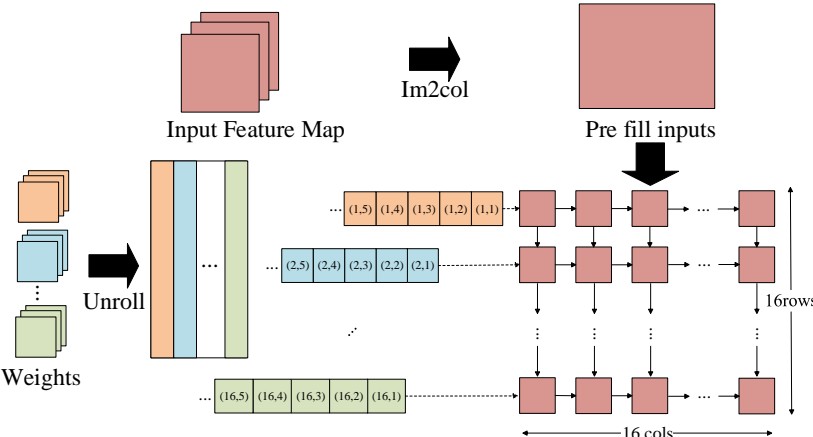

**Figure 4.** Example of simulating affine function. The example uses input stationary dataflow. If the workload is convolution, it is converted to GEMM by im2col.

### 5.2. Simulation of Tile Computation

Many previous works add idealized preconditions when modeling. For example, TENET assumes both that the accelerator has sufficient data during computation and that there is no dependence between the data. In the real computation process, however, these assumptions are not always true, meaning that the obtained simulation results are not accurate. To describe the hardware implementation in more detail, we comprehensively consider the adverse effects of data dependence, hardware resource constraints and other

factors on the computation in the simulation within the tile, and subsequently model the accelerator in a fine-grained manner.

More specifically, for a given tile, we perform secondary mapping based on the shape of the tile and the scale of the systolic array. Figure 5 illustrates this process. For the two matrices of m × k scale and k × n scale, we use the (m, x, x) mapping scheme to decompose them into multiple sub-tiles. Here, x represents the number of processing units on one side of the 2D systolic array. Since the computation process and sequence of the sub-tiles are fixed once the dataflow has been determined, we can easily find the time overhead of each sub-tile. It should be noted here that we calculated the time cost corresponding to each sub-tile, and then accumulated and summed it according to the constraints. When dealing with similar problems, Timeloop uses the computation of the first, second and final iterations of each loop; for irregular tile methods, however, this solution is not feasible. This is because there will be a large number of irregular sub-tiles at the edge of the tile, the shapes of which are often different from the first sub-tile, and their influence on the computation is also different.

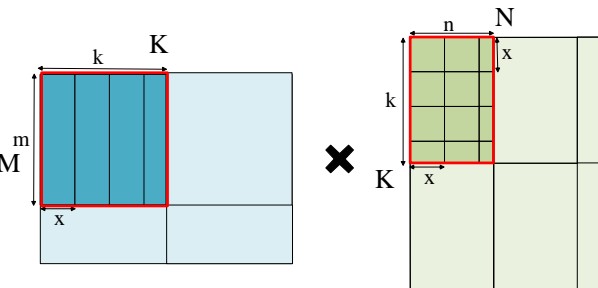

**Figure 5.** Mapping inside the tile.

Figure 6 presents an example of the internal simulation process of the tile. Let us take the process of input stationary dataflow as an example. For a given m × k matrix and k × n matrix, we map them to a systolic array of x × x scale for computation. We then partition the matrix according to the mapping method of (m, x, x), use col to indicate the number of columns in the sub-tile, and use row to indicate the number of rows in the sub-tile. On this basis, we first assume that the pipeline can operate normally and provide the corresponding computation time for each sub-tile. It should be noted here that the sub-tiles in the final row may be irregularly shaped, so we use conditional judgments here. We define s as the remainder of k divided by x. When s is 0, it indicates that k is divisible by x. At this point, the size of the last row of sub-tiles is the same as that of the previous row; otherwise, the size of the last row of sub-tiles is different, and the corresponding computation time will also change. Finally, we judge and simulate the possible bubbles in the pipeline. Since we adopted a double-buffer method in hardware implementation, we also simulate this when considering pipeline stalls. When we determine that bubbles will exist in the pipeline based on the present conditions, the computation time of the corresponding tile will also be extended.

This example illustrates our idea of simulating the internal computation process of the tile. For the output stationary dataflow and weight stationary dataflow, we can also conduct simulations in a similar way. In addition, this approach is highly parametric; we can modify the parameters to support the simulation of systolic arrays of different scales and computation delays.

```
1    // === The Simulation Process Inside The Tile ===
2    // Count the number of sub-tiles
3    col = ceil(n / x);
4    row = ceil(k / x);
5    s = k % x;
6    // Tile computation time when considering pipeline
7    for i = [1,col * row]:
8      if (i == 1):
9        time[i] = m + x * latency;
10     else if (i == col * (row - 1) + 1):
11       if (s != 0):
12         time[i] = m + s * latency + x − s;
13       else
14         time[i] = m;
15     else
16       time[i] = m;
17   // determine whether hardware resources are limited
18   for i = [3,,col * row]:
19     if (i % 2 == 1):
20       if (m < x * latency):
21         time[i] = x * latency;
22
23   for i = [1,col * row]:
24     totaltime += time[i];
```

**Figure 6.** The simulation process inside the tile.

### 5.3. Dependency Simulation

When we build the model, we also need to simulate the dependencies. Speaking generally, we consider the issue from two aspects: data dependence and hardware resource dependence.

When considering data dependency, it is necessary to consider the dependence of data between tiles. In our model, each tile is stored independently in an on-chip buffer, and the transmission order of the tiles depends on the dataflow. Therefore, once the dataflow is determined, we can accurately calculate the transmission order of different tiles. Under the premise of supporting the use of dual-buffer optimization in the hardware, we can also accurately determine the tiles that do not need to be repeatedly moved during the computation process, which further improves the accuracy of the simulation. On the other hand, when simulating computations within a single tile, we also consider data dependence. Figure 7 illustrates an example of data dependency within a tile. In the computation process, the a2 and b2 tiles need to take the computation results of the a1 and b1 tiles as input. At this point, the data-dependence relationship will cause the pipeline to be incontinuous, which substantially reduces the computational efficiency. It should be noted here that this kind of data dependence cannot be avoided through the use of dual buffers with the hardware; the resulting systolic array pause will have a significant negative impact on computing efficiency.

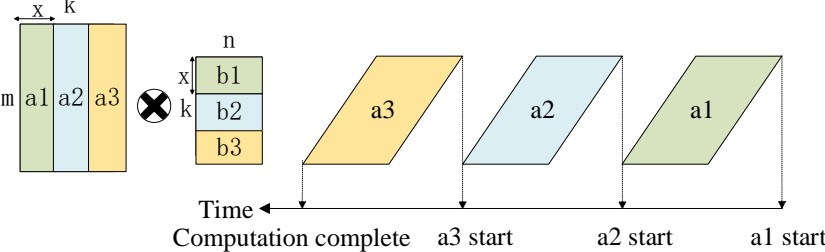

**Figure 7.** Example of data dependency within the tile.

On the subject of hardware resource dependency, we consider two aspects. We first need to consider the bandwidth resources that are occupied when the tile is transferred from the off-chip to the on-chip buffer. When the bandwidth resources are insufficient, the data transmission speed is slow. At this time, it may be the case that the original data of the on-chip buffer have been calculated but the new data have not yet been transmitted; this will lead to a significant reduction in computational efficiency. Moreover, due to the limited registers in the processing element of the systolic array, there may also be pauses in

the computation process within the tile, which is essentially a manifestation of hardware resource dependence.

## 6. FG-SIM Inputs and Mapper

### 6.1. Workload Specification

FG-SIM supports two workloads: one format is similar to the form of a single convolutional layer, and the other takes the form of matrix-and-matrix multiplication. Due to the different structural characteristics of different DNN network layers, the corresponding optimal mapping also differs. By simulating and exploring each layer of the network separately, we can obtain the optimal mapping of the DNN network as a whole, and accordingly achieve the highest computational efficiency. Figure 8 illustrates the description and expansion of the convolutional layer. These layers can be described as a 7D nested loop. Here, W0 and H0 represent the output tensor's height and width, WK and HK represent the weight tensor's height and width, W1 and H1 represent the input tensor's height and width, C1 represents the number of input channels, C0 represents the number of output channels, and B represents the number of inputs or batch size. As can be seen from Figure 8, for a given loop unrolling method, a specific computation process is determined. When we wish to divide data into tiles through mapping, we can calculate the order and scale of different tiles according to the index of the loop, thereby providing support for subsequent fine-grained modeling and dependency analysis. For the operation of the matrix-and-matrix multiplication, the simulator can also construct a mapping space according to its operation rules and search for the optimal solution.

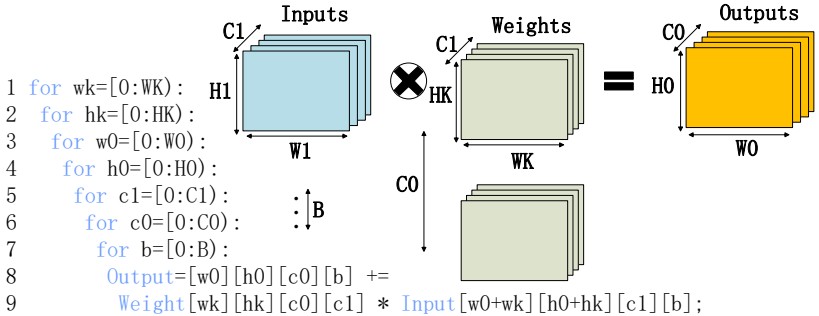

**Figure 8.** Convolutional layer 7D loop nest.

### 6.2. Mappings

Mapping describes the way in which data are partitioned into multiple tiles in the memory hierarchy. We discuss mapping here because the memory cannot store all data involved in the computation at the same time, while different mapping methods also have a huge impact on computing efficiency. Our simulator supports two workloads. For the convolution computation, we can convert it to a matrix-and-matrix multiplication operation via im2col. Therefore, in the following mapping-related discussion, we take matrix-and-matrix multiplication as an example.

When discussing the mapping scheme, we consider two aspects: the size of the tiles and the method of tile reuse. More specifically, different tile sizes have different impacts on computation time and data migration time. Only by considering the requirements of computation and data migration can we ensure that the computation time and data migration time overlap as much as possible, allowing us to finally find the most efficient mapping. On the other hand, there is more than one tile multiplexing method; each of these methods is limited by hardware storage resources and performs differently under different resource conditions.

Figure 9 shows two different approaches to mapping. The two matrices in the figure are of size M × K and K × N. We use tiles of size m1 × k1, k1 × n1, m2 × k2 and k2 × n2 to slice the matrices and (m1, k1, n1) and (m2, k2, n2) to denote the corresponding mapping

methods. Different segmentation methods will lead to irregular segmentation at the edge of the matrix, as reflected in the figure. In the process of exploring the optimal mapping, we seek to complete the computation as efficiently as possible. On the one hand, a larger tile method will cause the tile scoring computation time to exceed the data movement time, which helps to hide the data movement time in the computation time and further improves efficiency. On the other hand, a larger tile method often leads to irregular tile division of the matrix at the edge, which in turn causes the data movement time of these tiles to be higher than the computation time; a smaller tile method can largely avoid this specific problem. Therefore, the process of finding the optimal mapping essentially works to balance this contradiction as much as possible.

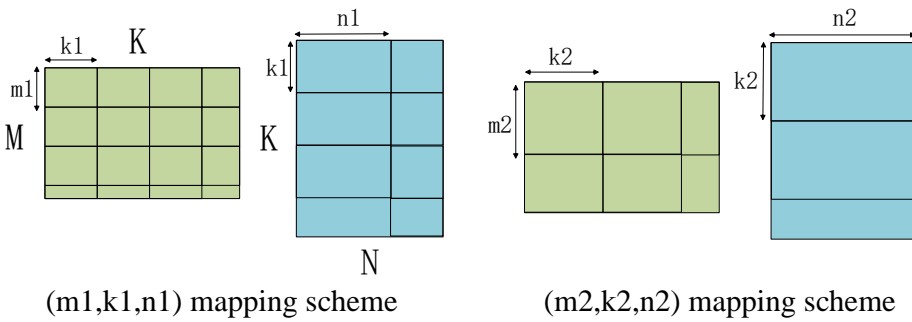

(m1,k1,n1) mapping scheme          (m2,k2,n2) mapping scheme

**Figure 9.** Examples of different mapping schemes.

Figure 10 illustrates two different tile multiplexing methods. The above method is a common tile method for matrix-and-matrix multiplication. The advantage of this method is that it can quickly calculate part of the final result and reduce the storage pressure of the intermediate result in the on-chip buffer. The disadvantage of this method is that it needs to repeatedly read the data of the original matrix; thus, we refer to this approach as result multiplexing. The method below is called process reuse. The advantage of this method is that there is no need to repeatedly access the original data involved in the operation; moreover, each tile of the original data only needs to enter the on-chip buffer once. The disadvantage of this method is that the intermediate result produced by the computation is the same size as the final result. Accordingly, if the on-chip buffer size is insufficient, the intermediate results will be repeatedly transmitted between off-chip and on-chip, which will reduce the overall computation efficiency.

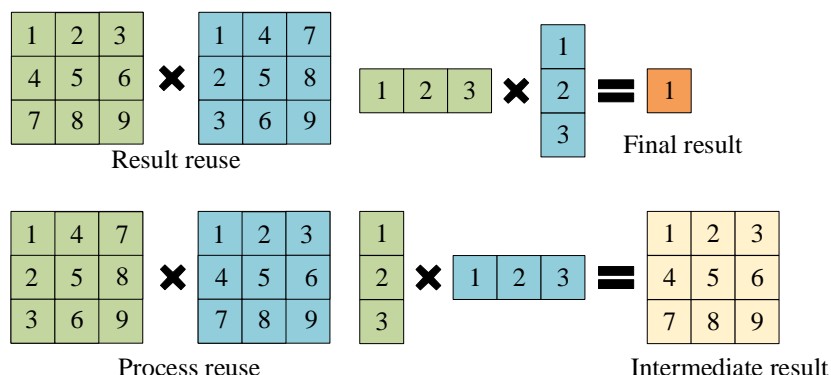

**Figure 10.** Examples of different tile multiplexing methods.

*6.3. Map Space Construction and Search*

When constructing the mapping space, due to the constraints imposed by the architecture and hardware resources, our mapping space is a subset of the complete mapping space. For example, our mapping method is limited by the on-chip buffer size, as the tile size cannot exceed it.

For a given workload, we comprehensively consider these constraints and then construct the corresponding mapping space. For example, for a matrix-and-matrix multiplication of M × K and K × N, we define (m, k, n) as a point in the mapping space. By changing the values of m, k, n, we can obtain a series of points that constitute the original mapping space. On this basis, we trim the mapping space through constraints such as hardware resource constraints, while the remaining mappings that meet the requirements constitute the real mapping space.

When we use random sampling-based heuristics to search, we randomly select m, k, n in the mapping space. By inputting M, K, N and (m, k, n) as parameters to the simulator, we can obtain the corresponding computation time. The result obtained by random fetching each time will be compared with the previously obtained optimal mapping scheme: if the effect of the new mapping scheme is better, the scheme and the corresponding optimal computation time will be updated. Finally, when the number of random fetches reaches a certain threshold, we stop exploring and arrive at the final result.

For small-scale workloads, such as the convolutional layer of a common DNN network, we typically use the traversal method to find the optimal mapping. For large-scale matrix-and-matrix multiplication, we use a random sampling-based heuristic to search. More sophisticated search methods will be presented in future work.

## 7. FG-SIM Validation

### 7.1. Experimental Setup

Due to the large number of free parameters and the diversity of workloads, many issues need to be considered when the aim is to ensure that a systolic array accelerator achieves optimal hardware performance. In order to take full advantage of the fine-grained modeling, we conducted investigations from the three perspectives of mapping methods, hardware resources and data reuse methods, then combined the above factors to identify the best computing solution.

In the next series of experiments, unless otherwise stated, we use a 16 × 16 scale systolic array in our experiments, with a single on-chip buffer size of 512 KB, and a single data size of 8 bytes. It should be emphasized here that we allocate separate on-chip buffers for the input, weights, and intermediate results and adopt the optimization of dual buffers.

We use (m, k, n, p) to define the computation scheme. Here, m, k, n represent the size of the tile, while the corresponding (m, k, n) represents the mapping scheme that was introduced above. In addition, we define p as the tile multiplexing method. When p is 0, this indicates that the result reuse is used, and when p is 1, this means process reuse.

### 7.2. Use Case 1: GEMM

FG-SIM supports general matrix multiplication. We conducted a GEMM test on two 500 × 500-scale matrices. Figure 11 plots the computation time, data movement time and total time corresponding to different computation schemes under the same hardware conditions. As can be seen from the figure, when we adopt a smaller tile method, the overall computation time will increase. There are many reasons why this occurs. When the on-chip buffer is sufficient, the smaller tile method will result in frequent data transmission between off-chip storage and the on-chip buffer, which increases the overall data transmission time. In addition, the data in the systolic array must be cleared and re-entered when switching the on-chip buffers. Therefore, a smaller tile method will result in frequent switching of on-chip buffers, which further increases the computational time cost. Finally, the smaller tile method will also cause bubbles in the pipeline, which further reduces the computation efficiency.

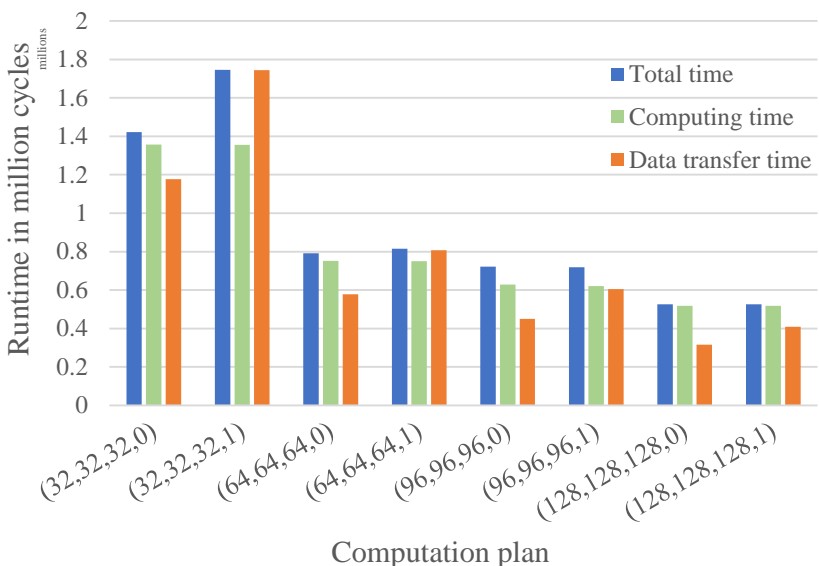

**Figure 11.** The time cost of GEMM under different computation schemes.

### 7.3. Use Case 2: Convolution

FG-SIM supports general convolution computations. We test the first layer of convolution of the Alexnet network and search for the optimal computation scheme for different stride conditions. Figure 12 plots the computation time, data movement time and total time of the optimal computation scheme corresponding to different strides. In addition, we modify the size of the convolution kernel based on common convolutional networks. Table 2 lists the best computation scheme for different convolution kernels and different strides. By modifying the input image, the stride and the size of the convolution kernel, we are able to simulate any convolution computation. As can be seen from the table, for convolutional networks, different convolutional layers have different characteristics, while the corresponding optimal computation schemes are also different. Therefore, in cases where hardware resources are fixed, exploring the best computation scheme for different convolutional layers also helps to improve the overall computation efficiency.

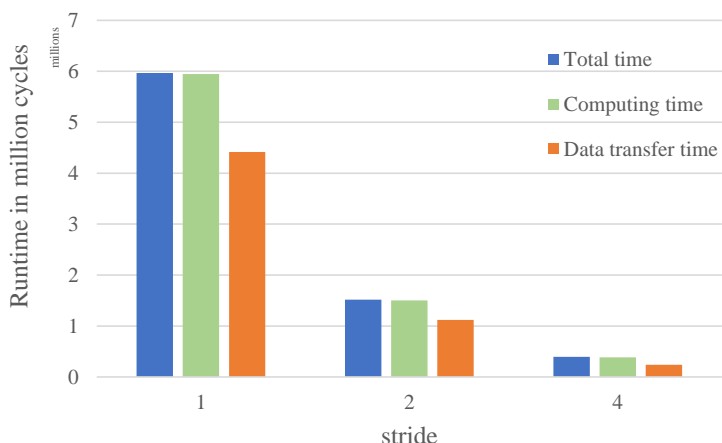

**Figure 12.** The time cost of convolution under different strides.

**Table 2.** The optimal computation scheme of convolution under different convolution kernels and different strides.

| Kernel Size | Stride | | |
|:---:|:---:|:---:|:---:|
| | 1 | 2 | 4 |
| $3 \times 3$ | (64, 16, 992, 0) | (64, 16, 912, 0) | (64, 16, 800, 0) |
| $5 \times 5$ | (64, 40, 992, 0) | (64, 40, 544, 0) | (64, 40, 256, 0) |
| $7 \times 7$ | (64, 80, 800, 0) | (64, 80, 352, 0) | (64, 80, 192, 0) |
| $11 \times 11$ | (64, 128, 480, 0) | (64, 192, 224, 0) | (64, 192, 128, 1) |

*7.4. Use Case 3: PE Utilization*

FG-SIM estimates the PE utilization accurately, which is another key factor that affects the latency. We compare the PE utilization calculated by our work and TENET using the dataflow proposed by Scale-sim for AlexNet. We use the reported PE utilization in Scale-sim as the golden result. Figure 13 shows the PE utilization obtained by three simulators simulating each convolutional layer of the AlexNet network under the condition of an $8 \times 8$ systolic array when using weight-stationary dataflow. It should be pointed out that FG-SIM uses the results of the optimal mapping scheme. Overall, FG-SIM improves the estimation accuracy of PE utilization from 92.4% to 99.5%. This improvement is partly attributed to the fine-grained modeling method of FG-SIM, which can more accurately characterize the scenarios where PE flexibly uses data in different registers during computation. When calculating the average PE utilization, TENET is only sensitive to some parameters. Its approach is more similar to running individual simulations of a single-layer loop and computing the average. This simulation method ignores the fact that the systolic array is still being computed during the unrolling of the multilayer loop, so its estimate will be lower. When using weight-stationary data flow, TENET is only sensitive to the size of the output matrix. For the C1 and C2 convolutional layers with large output matrices, the estimated results are more accurate at this time, while for C3, C4 and C5 with small output matrices, the estimated accuracy of PE utilization is greatly reduced, and the results are the same.

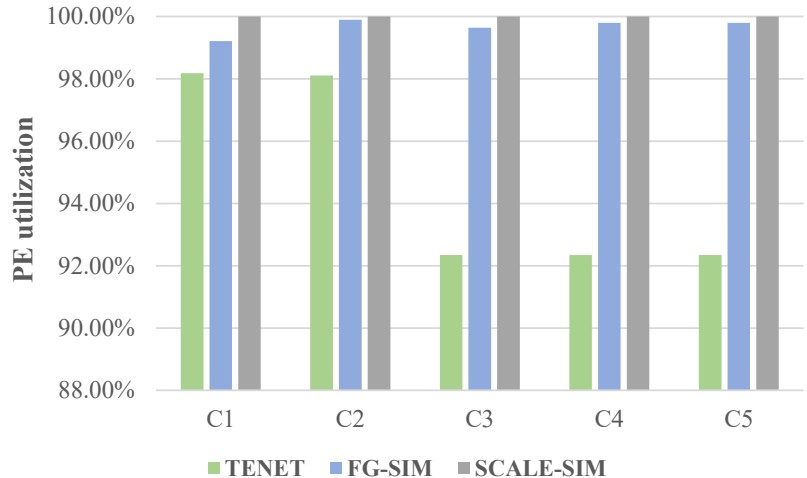

**Figure 13.** PE utilization comparison with TENET and Scale-sim on AlexNet, C1–C5 means CONV1, CONV2, . . . , CONV5.

*7.5. Use Case 4: Buffer Size*

The previous experiments were all carried out under the same hardware resources. When we change the size of the on-chip buffer, the optimal computation scheme will also be both affected and changed. We choose the third convolutional layer of the Alexnet network for testing. As the on-chip buffer gradually increases in size, the total time spent on the

computation of the convolutional layer continues to decrease. This reflects the impact of hardware resources on computing efficiency.

In addition, we also note that as the size of the on-chip buffer gradually increases, its effect on computing performance is gradually reduced, which is a manifestation of marginal effects. Figure 14 plots the optimal computation time and the corresponding buffer utilization under different buffer sizes. When the buffer is small, the buffer utilization is higher. At this point, increasing the buffer size will significantly improve the computing performance. As the buffer gradually increases, the utilization of the buffer size gradually decreases, and the improvement in computing performance becomes smaller and smaller. We contend that this evaluation method can provide hardware designers with the support required to adjust the hardware structure and allocate the related resources.

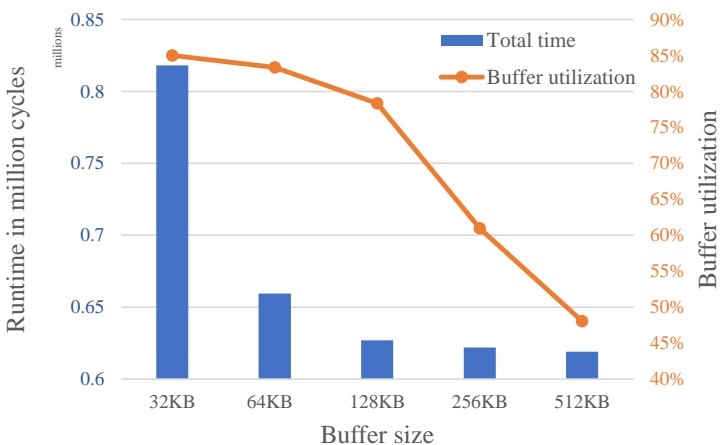

**Figure 14.** Optimal computation time and buffer utilization of convolution under different buffer sizes.

### 7.6. Validation Results

We tested common convolutional networks on TENET, FG-SIM and a hardware emulation platform. Since TENET only gives the computation result for the best mapping, in order to ensure that the results are objective, we also use the best mapping for comparison. We take the test time of the hardware emulator as the benchmark and divide the simulation time of TENET and FG-SIM by the benchmark time to obtain the accuracy. Figure 15 presents the test results of five common convolutional networks.

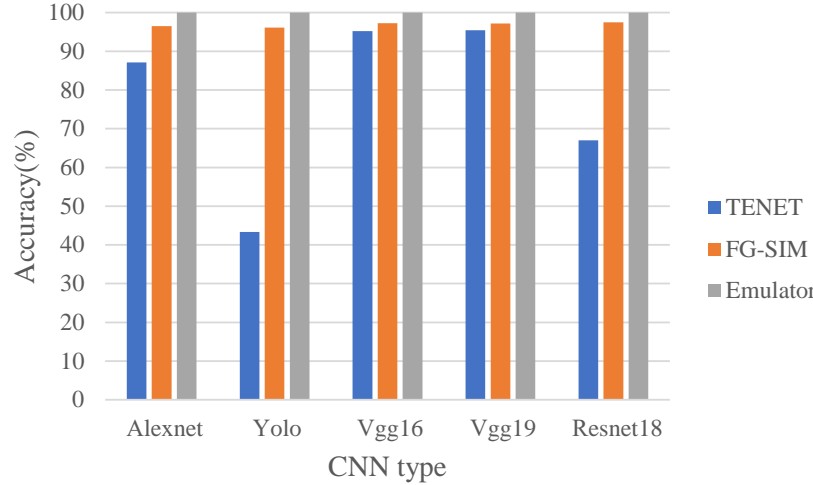

**Figure 15.** Comparison of TENET, FG-SIM and emulator under five convolutional networks.

As can be seen from Figure 15, the result of FG-SIM is very close to the time of the hardware simulation platform; in fact, its accuracy exceeds 95%. By contrast, the accuracy of

TENET fluctuates greatly. This phenomenon occurs because the characteristics of different CNN convolutional layers are different. Because the W0 and H0 of the convolutional layer are small, it is difficult for Yolo [27], Resnet18 [28] and Alexnet [29] to ensure sufficient data supply during the tile computation process. This will cause many pauses in the computation of the systolic array, which reduces the overall computational efficiency. When TENET built the model, it was assumed that no correlation existed between the data points and that the data supply was sufficient. Therefore, when there is a problem with this assumption, the accuracy of TENET will be greatly reduced.

## 8. Conclusions

In this paper, we introduced FG-SIM, a fine-grained modeling method for evaluating systolic array accelerators. It disproves the ideal assumptions made in modeling and describes the specifics of the hardware implementation. Overall, FG-SIM is a good tool for exploring the microarchitecture under a systolic array structure. It can provide more accurate simulation results and provide specific guidance on the complex trade-offs of hardware design.

**Author Contributions:** Writing—original draft, Y.L. and M.W.; Writing—review and editing, Y.L., M.W., J.F., J.S. and Y.C.; Figures, Y.L.; Study design, J.F.; The other work, all authors contribute equally. All authors have read and agreed to the published version of the manuscript.

**Funding:** This research was funded by National Nature Science Foundation of China under NSFC Nos. 61802420 and 62002366.

**Institutional Review Board Statement:** Not applicable.

**Informed Consent Statement:** Not applicable.

**Data Availability Statement:** Not applicable.

**Conflicts of Interest:** The authors declare no conflict of interest.

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
