# Peer review of "A Fine-Grained Modeling Approach for Systolic Array-Based Accelerator"

_electronics, doi:10.3390/electronics11182928_

Round 1
Reviewer 1 Report
Comment 1: The authors need to explain more precisely why the problem addressed in the manuscript is important.
Comment 2: The following sentences should be re-written.
-Deploying hardware accelerators to accelerate training is a method commonly employed to solve such problems.
-However, whether due to issues of generality or abstraction, it is difficult for these works to fully describe the specific implementation details of the hardware.
-In these works, in order to maximize versatility, they need to describe the organization of different designs and find an effective mapping in the generated mapping space for evaluation.
Comment 3: It is not common to place Figures in the Introduction. This should be reorganized.
Comment 4: Is it possible to quantify the complexity of the proposed approach/method/algorithm?
Comment 5: Conclusion part of the manuscript sounds like a summary. Brief conclusions are therefore necessary.
Author Response
Dear reviewer, according to your review comments, I have made corresponding revisions to my manuscript, and I will explain item by item below.
For Comment 1, I made a further explanation in the second paragraph of the introduction, and elaborated the adverse effects of the problem in more detail to illustrate the importance of the problems mentioned in the article.
For Comment 2, I found the corresponding sentence in the article and rewritten its content. You can check the details in the revised version of the manuscript.
For Comment 3, I moved the structure diagram from introduction to chapter 4 and added a more detailed description.
For Comment 4, it is difficult to quantify the complexity of the approach because my proposed method is closely related to the workload, hardware resources, and dataflow, and there are also inter-constrained correlations among these factors.
For Comment 5, I wrap up the conclusion section to get a brief conclusion.

Reviewer 2 Report
“A Fine-grained Modeling Approach for Systolic Array-based Accelerator” is nicely written report on new tool FG-SIM. It has extensive list of references, even though the bulk of them are rather old. Since it is rather report then its scientific merit is hard to characterize. Perhaps it would have been also good to define exactly what this FG-SIM means, even if it can be guessed that we are talking about fine-grained simulator. Perhaps Introduction, motivation and related work could be somewhat rearranged, as the proposed solution appears before the discussion of similar works and before it is defined why it is beneficial. Also, some concepts like systolic array could be perhaps briefly introduced. Relationship between figures 1 and 3 could be better explained. Is the caption under figure 14, correct?
Author Response
Dear reviewer, according to your review comments, I have made corresponding revisions to my manuscript, and I will explain item by item below.
- I mentioned the basic definition of FG-SIM in the abstract and rewrote the third paragraph of the introduction to explain it in more detail.
- I revised the second and third paragraphs of the introduction. The second paragraph briefly introduces the inadequacies of the related work and marks the citations. The third paragraph introduces the basic situation and advantages of the proposed method. This ensures that the proposed scheme follows discussions of similar work.
- I moved Figure 1 from the introduction to Chapter 4, and added an introduction to systolic array-related concepts, as well as an explanation of the connection between Figures 1 and 3.
- I have revised the title of Figure 14, which was indeed a typo before, thank you very much for pointing it out.

Round 2
Reviewer 1 Report
I am pleased with the way the authors have revised the manuscript in line with the comments I have given.